# Developmental Toxicity Studies: The Path towards Humanized 3D Stem Cell-Based Models

**DOI:** 10.3390/ijms24054857

**Published:** 2023-03-02

**Authors:** Mariana A. Branco, Tiago C. Nunes, Joaquim M. S. Cabral, Maria Margarida Diogo

**Affiliations:** 1Collaborative Laboratory to Foster Translation and Drug Discovery, Accelbio, 3030-197 Cantanhede, Portugal; 2IBB—Institute for Bioengineering and Biosciences, Department of Bioengineering Instituto Superior Técnico, Universidade de Lisboa, 1049-001 Lisboa, Portugal; 3Associate Laboratory i4HB—Institute for Health and Bioeconomy, Instituto Superior Técnico, Universidade de Lisboa, 1049-001 Lisboa, Portugal

**Keywords:** developmental toxicity, human stem cell-based teratogenic models, in vitro human teratogenicity, human heart organoids, human gastruloids

## Abstract

Today, it is recognized that medicines will eventually be needed during pregnancy to help prevent to, ameliorate or treat an illness, either due to gestation-related medical conditions or pre-existing diseases. Adding to that, the rate of drug prescription to pregnant women has increased over the past few years, in accordance with the increasing trend to postpone childbirth to a later age. However, in spite of these trends, information regarding teratogenic risk in humans is often missing for most of the purchased drugs. So far, animal models have been the gold standard to obtain teratogenic data, but inter-species differences have limited the suitability of those models to predict human-specific outcomes, contributing to misidentified human teratogenicity. Therefore, the development of physiologically relevant in vitro humanized models can be the key to surpassing this limitation. In this context, this review describes the pathway towards the introduction of human pluripotent stem cell-derived models in developmental toxicity studies. Moreover, as an illustration of their relevance, a particular emphasis will be placed on those models that recapitulate two very important early developmental stages, namely gastrulation and cardiac specification.

## 1. Introduction

Developmental toxicity accounts for the impaired embryonic/fetal development of an organism due to exposure to substances. The idea that the placenta forms an impenetrable barrier against most drugs is now widely accepted as false. Evidence from the last decades has proven that a wide range of administrated drugs and environmental chemicals that pregnant women are exposed to can cross the placenta and enter the circulation of the developing embryo. This has exacerbated the need to carefully consider drug administration during pregnancy and to promote the development of studies to specifically account for the teratogenic effects of newly developed or already existent drugs, primarily in the cases of pharmaceuticals that are intended for or need to be used in women of reproductive age or during pregnancy [1].

Information regarding teratogenic risk in humans is often missing for most of drugs purchased. Within the group of drugs approved by the FDA from 1980 to 2010, less than 10% have sufficient pregnancy-related data to determine fetal risk [2]. Additionally, a review of approved drugs between 2003 and 2012 revealed that most pharmaceuticals had pregnancy data based on animal studies and only 5.2% had human pregnancy data [3]. In fact, in the US, between 2011 and 2012, 95% of the ongoing clinical trials excluded pregnant women and only 1% were designed specifically for this population group [4,5].

Either due to pregnancy-related medical conditions or pre-existing diseases, medicines are eventually needed during pregnancy to help prevent, ameliorate or treat an illness. Over the past few years, there it has been an increased rate of medication used during pregnancy [6]. In 2015, almost half of pregnant women were using four or more drugs at some point during pregnancy, this being predominantly registered in the first semester, a crucial period for organogenesis. An additional point of concern is related with the fact that 51% of pregnancies are unplanned [7], which can lead to unintentional drug exposure. In fact, for certain drugs, stopping treatment does not translate into immediate termination of potential teratogenicity due to the delayed elimination of such substances from the plasma [8]. 

The lack of human teratogenicity data for most approved drugs has driven doctors to base their therapeutic strategies for pregnant women on cohort studies that account for collected healthcare records and clinical metadata analyses that assess the link between teratogenesis and drug exposure during human pregnancy in retrospect. These studies revealed an increased administration rate of some subtypes of drugs, including antibiotics [9], antidepressants and anticonvulsants [10], over recent years. Reports suggest that antibiotics account for nearly 80% of all prescribed medication during pregnancy and that approximately one in four women will receive an antibiotic during pregnancy [11]. In fact, there are common infections that occur during pregnancy, such as urinary and upper respiratory tract infections, that, if not treated, can be associated with significant morbidity and spontaneous abortion. Additionally, the use of antidepressants during pregnancy has increased over the past years, with 6–8% of pregnant women being prescribed or using an antidepressant [12,13]. Although data linking antidepressants to teratogenicity, particularly to heart defects, are still controversial, different clinical retrospective studies, based on live birth cases, have pointed to an increased risk of congenital malformations [13,14,15]. More information is required about drugs that are used in the treatment of conditions that may not be interrupted during pregnancy, such as in the case of anti-cancer treatments or the case of lifelong conditions, such as multiple sclerosis [8], epilepsy and autoimmune diseases, including lupus, which have also been identified as medical disorders that require adapted therapeutic strategies to ensure both maternal and fetal safety. Although the rate of cases where pregnant women are involved in cancer treatments is still low [16,17], the increasing trend to postpone childbirth until a later age is expected to induce an increment of these cases [6]. Among the lifelong medical conditions, epilepsy is one of the most frequent neurologic disorders in pregnancy, with a prevalence of 0.3–0.5% [18], and fetal exposure to anti-epileptic drugs results in abnormal development in 2.2–11% of the cases [19]. Common pregnancy-related medical conditions, such as hypertensive disorder, which affects about 10% of all pregnancies [20], remain amongst the leading causes of maternal and perinatal morbidity and mortality. Therefore, it is clear that for a significant number of drug subclasses, data regarding teratogenic defects is urgent. Apart from drug administration, environmental pollution exposure during pregnancy has been also considered as a critical point that should be carefully accounted for. In fact, maternal exposure to metals, chemicals, and toxins and their link to congenital anomalies is well documented (reviewed in [21]), with the World Health Organization estimating that 5% (2–10%) of all congenital anomalies are attributable to environmental causes [22]. As the heart is the first functional organ to be formed, understanding the impact of substance exposure on heart organogenesis is often used as a critical parameter for developmental toxicity prediction in vivo. In fact, severe cardiovascular dysfunction is lethal for embryos with approximately 3–4 weeks of gestation [23]. Due to this, it is often difficult to identify cardiac developmental toxicity since most of the retrospective clinical data account only for defects observed after birth, which only capture a small window of the possible impact that a drug can have during embryonic development. Drugs that cause a lethal malformation can be therefore misread. Excluding pregnancies ending in terminations due to live-threat early heart defects, congenital heart disease (CHD) is the most common congenital abnormality worldwide, affecting 8 to 12 infants per 1000 births globally. However, its causes remain mainly unknown, with only up to 15% of CHD cases having a determined genetic cause [21]. Nongenetic causes for CHD encompass drug exposure, maternal illnesses, and dietary behaviours.

Although retrospective clinical data have given import insights and created awareness regarding the impact of drug administration during pregnancy and the possible link with teratogenicity, the future depends on the development of models that can predict this developmental toxicity before putting the fetus and the mother at risk. This review summarizes the most commonly used animal in vivo and in vitro models to assess developmental toxicity, with a special emphasis placed on human pluripotent stem cell (hPSC)-derived models recapitulating the early stages of gastrulation and cardiac specification.

## 2. Animal Models in Developmental Toxicity Assessment

Every year, pharmaceutic companies use tens of thousands of animals for toxicology tests, which obviously raises ethical concerns and the awareness of the need to implement alternative in vitro models to minimize the use of animals. However, these models are still the leading pre-clinical strategy for assessing the toxicity of new developed drugs. Among mammalian research models used in developmental toxicity studies, the mouse model is one of the most widely used, since mice are small animal and present relatively similar sequence of embryonic development compared to humans [24]. However, this model presents drawbacks including the fact that mouse embryos are difficult to access inside the uterus, preventing non-invasive in vivo imaging of early embryonic stages [25] and offering low-throughput drug toxicity screening. Within non-mammalian organisms, including zebrafish (Danio rerio), fruit fly (Drosophila melanogaster), frog (Xenopus laevis), and chicken (Gallus gallus) [26], zebrafish has unique characteristics for developmental toxicity studies. The fact that embryos develop externally and have a small size allows chemical compounds to be directly diluted into the water and to diffuse into embryos. Early optical transparency also gives the opportunity to optically monitor the dynamic cellular events of embryogenesis [27]. Overall, these features make this a high-throughput and low-cost model organism, with advantages over other animal models. However, zebrafish is evolutionarily distinct from humans, presenting some developmental differences, for example in heart organogenesis [26].

## 3. In Vitro Animal-Based Models to Assess Teratogenicity

Three in vitro teratogenicity testing platforms have been validated by the EU Reference Laboratory for Alternatives to Animal Testing (EURL-ECVAM). These include (1) the whole-embryo culture (WEC), (2) the limb bud micromass culture, and (3) the mouse embryonic stem cell test (mEST) [28]. The whole embryo culture uses explanted rodent embryos at the 1–5 somite stage, the early organogenesis stage, that are prepared with the visceral yolk sac and ectoplacental cone intact. Embryos are cultured in roller bottles for 48 h, although cultures can be extended for up to 72 h. At the beginning of the culture, test compounds are added to the culture medium at various concentrations and embryos are examined for viability, a factor determined by heartbeat and yolk sac circulation and growth, as measured by crown–rump length and/or protein content and morphology. The rate of malformation and of the occurrence dead embryos of each test group can be used to determine parameters such as the half-maximal inhibitory concentration (IC50) [28,29]. A primary advantage of this model includes the fact that embryos are easily accessible and manipulated during culture. However, embryos can only be cultured for a short period of time, which may be a limitation to this method. In fact, although this window of time represents a critical step in organogenesis and thus sensitivity to teratogens, the effects may not manifest morphologically until a later time in embryonic development. Another limitation is that this model still requires animal sacrifice [29]. The micromass culture is based on high-density culture of embryonic limb bud mesenchymal cells, from chick, mouse, or rat specimens. This model has the advantage of not requiring the sacrifice of the mother to access the embryo [30], but the studies are restricted to assessing the early stages of skeletal development. The mEST is the only test of the three in vitro alternative tests validated by ECVAM that does not require live laboratory animals. The mEST is based on the 3D spontaneous differentiation of mouse embryonic stem cells (mESCs) for 10 days, a system known as embryoid body (EB) differentiation, with continuous drug exposure. The cells tend to differentiate into cardiomyocytes under these conditions, and the percentage of beating EBs are scored microscopically. From the results of this test, the drugs are classified as not embryotoxic, weakly embryotoxic, or strongly embryotoxic, compared to the toxic concentrations observed in adult fibroblasts. Although this model does not rely on live animals and uses only commercially available cell lines, the qualitative nature of scoring “beating” EBs, which requires experience and is subjected to observer bias, has driven scientists to propose modifications to this test, including alteration of the assessed readout at the endpoint of culture to more quantitative parameters and the inclusion of additional intermediate endpoints.

Although animals or animal-based in vitro models have been essential for toxicology testing over the past years, species differences between humans and animals have been responsible for the lack of drug-induced teratogenicity detection. The most known and recognized case is the “Thalidomide Tragedy”, in which the teratogenicity of this drug was not foreseen in mouse models, leading to numerous embryonic, fetal and neonatal deaths, and severe congenital malformations in humans. This was one of the first demonstrations that animal studies do not obviate concerns related to teratogens that are uniquely harmful to human fetal development. In fact, it has been reported that only 70–80% of the teratogenic cases observed in rats and rabbits are reflected in humans [31,32], and as a result 9 in 10 drugs that enter human clinical trials fail because they are unsafe or ineffective [33], reinforcing the idea that there is a need to change the way pre-clinical tests are being conducted.

## 4. In Vitro Humanized Models to Assess Developmental Toxicity

In recent years, researchers have been focused on developing alternative in vitro human models to address the potential teratogenicity of drugs. Among other applications, hPSC-derived models offer the opportunity to complement the data obtained from animal developmental toxicological studies since hPSCs differentiation can mimic key aspects of human embryonic development in vitro. Although the topic has still not been extensively explored, there have been hPSC-based studies focused on assessing the impact of drug exposure during the gastrulation-like stage or specifically in the mesendoderm/cardiac differentiation described in the literature. The focus of these studies has been not only the identification of the potential teratogenicity of a specific drug and the regulatory mechanisms behind the observed toxicity, but also the establishment of platforms exhibiting a high-throughput screening capacity while maintaining a sufficiently predictive value, which could be extremely helpful when it is necessary to perform the fast screening of several drugs.

Among the hPSC-derived models already applied for embryonic toxicity testing, the next sections will focus on (1) the 3D EBs and gastruloid models that have been implemented to assess the impairment of germ layers specification upon drug exposure and (2) the use of 2D/3D cardiac models to study heart developmental toxicity.

## 5. Multilineage Developmental Toxicity Studies—Use of HPSC-Derived EBs and Gastruloids

One of the firstly and most widely used human stem cell-based tests for predicting developmental toxicity in vitro relied on the use of EBs [34,35,36,37,38,39,40], and, more recently, on gastruloids [41], which allow researchers to study early developmental patterning mechanisms regulating the multilineage fates specification (Table 1). Unlike EBs, which are 3D models of spontaneous and unbiased differentiation of PSCs that do not present any in vivo-like structural organization, gastruloids display self-organized gene expression domains, resembling the anteroposterior axial elongation observed during embryonic development. Although EBs reflect cellular differentiation into all three germ layers, and therefore are relevant models for use to study the toxic effect of compounds at the blastocyst stage, gastruloids, being model systems that mimic some of the events of gastrulation, including symmetry break and axial elongation, have the additional asset of potentially giving information regarding spatial organization defects.

The impact of drug exposure in normal hPSC-derived EBs and gastruloid patterning has been mainly disclosed by performing bulk and single-cell RNA-sequencing (RNA-seq) methodologies. Particularly, bulk RNA-seq data analysis has been used to characterize the impact of valproic acid [35,36], thalidomide [37], folic acid, all-trans retinoic acid, and dexamethasone [36], after 14/15 days of continuous drug exposure during EB differentiation. Through a comparison of the differentially expressed genes in normal and drug-exposed conditions, the degree of drug-interference with normal developmental is quantified [35]. Gene ontology analysis of the differentially expressed genes in drug-exposed conditions confirmed the presence of previously in vivo identified congenital defects induced by those drugs, particularly, the association of valproic acid to nervous system development repression [35,36]; of thalidomide in heart, limb, vasculature and skeletal system development impairment [37]; and of retinoic acid in compromised primitive streak formation [36]. These studies demonstrated that RNA-seq analysis is a viable option for use to detect disrupted signals upon drug exposure, particularly to help deciphering the main differences behind abnormal commitment of PSCs into the three germ layers, helping in this way to validate data previously known from animal models in a human context while finding out new mechanistic avenues. Combining RNA-seq information from more than one drug could also allow the identification of mechanisms of action that are unique to a certain drug, or of common pathways that are involved in teratogenicity in general and are not restricted to a certain compound. Konala and co-workers, who assessed the teratogenic potential of folic acid, all-trans retinoic acid, dexamethasone, and valproic acid on the normal EB commitment, demonstrated that 5.3% of the dysregulated genes were common across all four drugs. This may indicate the existence of a set of genes that are commonly involved in teratogenicity. Despite that, they also identified gene clusters that were restrictedly dysregulated for a specific drug, identifying folic acid-treated conditions as the closest to the control group, and on the opposite side, retinoic acid treatment as the one that induces the highest level of developmental dysregulation [36]. However, with EBs being a heterogeneous population of cells, introducing single-cell RNA-seq may reveal additional information regarding less represented cell populations, which are diluted in bulk transcriptomics. Guo and colleagues used single-cell transcriptomic analysis to assess the impact of nicotine exposure on normal EB patterning after 21 days of differentiation [39]. The obtained data allowed them to identify six major types of progenitors and discriminate the impact of nicotine exposure on those sub-populations of cells. This allowed the researchers, for example, to identify that nicotine increased the propensity for arrhythmic Ca^2+^ release in hPSC-derived CMs. Additionally, the calculation of cell-cycle phase scores based on canonical markers identified that nicotine exposure specifically affected the cell cycle of endothelial, stromal, and muscle progenitor cells.

Although the advantages of using RNA-seq analysis to deeply understand the pathways behind the observed developmental toxicity are irrefutable, this methodology is not compatible with high-throughput screening settings. The development of screening platforms that allow researchers to detect, in a fast and reliable way, if a compound can be teratogenic or not, is also worthy of attention in this field. The selection of an easily accessible redout linked with machine learning algorithms has been explored to establish screening platforms for teratogenicity assessment. By using EB differentiation, Jaklin and colleagues selected a panel of eight drugs with available data regarding their positive or negative teratogenicity in humans and assessed the expression of a pre-selected battery of 96 developmentally related genes [38]. They found out that this panel of genes allows them to distinguish between teratogens and non-teratogens in concentrations below cytotoxicity. Linear statistical modeling identified 19 genes, from the initial 96, that were significantly regulated by teratogens and were sufficient to allow the accurate prediction of teratogenicity of seven out of the eight selected compounds. Although this study represents only a proof-of-concept with an initial validation set of compounds, future improvement of the machine-learning model with additional data from different classes of drugs, can be considered as a promising developmental toxicity prediction tool.

The inclusion of reporter cell lines in high-throughput screening settings has been also used as a viable option for in situ assessment of developmental toxicity. Mantziou and colleagues used a gastruloid model and the multiplex reporter cell line RUES2- GLR (mCit—*SOX2*, mCerulean—*BRA*, tdTomato—*SOX17*) [41], to follow the impact of 7 drugs with known teratogenicity on the commitment towards the three germ layers, mesoderm, endoderm and ectoderm, as analyzed through the expression of BRA, SOX17 and SOX2, respectively. With self-organization in polarized domains being amongst the unique features of gastruloids, the same authors focused on understanding how drug exposure can compromise axial elongation and the right anterior–posterior polarization of the different subpopulations of progenitor cells. To do so, the elongation distortion index and gastruloid size were assessed. Mantziou and colleagues proposed that, in a gastruloid setting, a close examination of gastruloid elongation can give a quick metric output with which to identify morphological defects following compound exposure. This follows the same mindset of defining a panel of genes to predict, in a reliable way, the compromised germ lineages specification.

## 6. Cardiac Developmental Toxicity Studies—Use of HPSC-Derived 2D and 3D Models

In vitro screening of heart developmental toxicity has been performed using different 2D and, more recently, 3D models based on hPSC mesendoderm [43,44,45] and CM differentiation [31,40,46,47,48,49,50,51,52,53,54] (Table 2). By applying different time windows of drug exposure, correlated with drug administration at different stages of pregnancy, and readout(s) assessment at the end of culture and at additional intermediate time points along differentiation, it has been possible to obtain information regarding the specific stages of cardiac commitment that are predominantly affected upon drug exposure.

Platforms based on 2D differentiation had already proved to be viable options to use to easily perform screening of several drugs in the context of developmental toxicity settings. As they were articularly interested in understanding developmental toxicity at the level of mesendoderm specification, Xing and colleagues cultured hPSC as 2D confined colonies by using micropatterned culture surfaces. These authors were capable of inducing reproducible spatial patterning of mesendoderm differentiation and also of recapitulating the cell migration processes observed during embryogenesis [44]. Data collected from a panel of 30 drug-like compounds allowed the authors to define, among other tested parameters, four morphological features based on Brachyury (T) protein expression (Table 2). These allowed them to quantitatively characterize the success of mesendoderm patterning and consequently allowed them to predict developmental toxicity [45]. Although this quantitative morphometric assay allows researchers to capture compound’s teratogenic effects over both cell differentiation and cell migration, achieving 100% specificity, 93% sensitivity and 97% accuracy in the 30 screened compounds with known teratogenic effect, 1 in 15 teratogens was classified still as a false negative. Nevertheless, by performing in parallel the same analysis using mESCs, the developed platform exhibited better selectivity for human-specific effects than the mEST. At the level of CM differentiation, Jamalpoor and colleagues also reported a rapid and reliable developmental toxicity screening assay, based on a direct 2D CM differentiation protocol [50]. The authors defined two readouts, namely the aggregate contraction rate, assessed at the end of differentiation by using a visual contraction score, and the expression of BMP4 as a mesoderm marker at D7 and MYH6 as a cardiomyocyte marker at D14. The predictivity of the platform, as accounted for the two defined readouts, was firstly validated with drugs that have a known impact on embryonic development, including thalidomide and valproic acid, being further tested with an additional 21 compounds. Compared with the teratogenic drug classification based on in vivo animal studies, the assay achieved high accuracy (91%), sensitivity (91%), and specificity (90%).

As already mentioned, differences in drug exposure timing have been also explored to understand if there is a critical phase of cardiac commitment that could be particularly sensitive to a specific drug. Liu and co-workers studied the cardiac developmental toxicity of 13-*cis*-retinoic acid using a 2D CM differentiation protocol. This was by testing two different time windows of drug exposure, namely until mesoderm specification or only during cardiac progenitor cells commitment and concluded that the toxicity of the studied drug was only reflected early on, at the level of mesoderm formation. Specifically, using RNA-seq data analysis, they concluded that drug exposure caused the dysregulation of genes involved in signaling pathways underlying mesoderm differentiation via HNF1B, SOX10 and NFIC, leading to disruption in mesoderm formation. To confirm this result, they tested the exposure of the drug after mesoderm formation and no perturbations in cardiac differentiation were observed. Additionally, they also demonstrated that drug exposure to ectoderm and endoderm formation had minor and moderate effects, respectively, pointing to a preferential mesoderm-specific defect upon drug exposure. On the opposite side, Ye and colleagues, who tested the toxicity of the antiviral drug Ribavirin at three different phases of differentiation (from pluripotent status to mesoderm, then to cardiac progenitor cells, and finally to CMs), concluded that the drug inhibited proliferation and differentiation in the last two stages but not in the first one, suggesting that ribavirin may be detrimental for cardiac proliferation and differentiation during the mid and late stages of differentiation [49]. Fu and colleagues studied the impact of the organic pollutant dioxin on cardiac commitment also by drug exposure at different time windows of differentiation [48]. Particularly, they analyzed the impact of drug exposure at pluripotency (D-3 to DO), mesoderm (D1 to D4), cardiac mesoderm (D5 to D8) and cardiac progenitor (D9 to D14) stages through evaluation of the CM markers ACTN2, TNNT2, and MYL2 expression at the end of the differentiation process. They found that drug treatment at the pluripotency or the mesoderm stages reduced CM differentiation efficiency, with the treatment at the pluripotency stage showing the strongest effect. In contrast, treatment at the cardiac mesoderm and cardiac progenitor stages showed no inhibition, meaning that dioxin inhibits CM differentiation by interfering with the commitment to the mesoderm and cardio-mesoderm lineages. They further identified that the effect was largely mediated by the aryl hydrocarbon receptor (AHR), and to corroborate this theory they treated human PSCs with the AHR antagonist CH223191 in conjunction with dioxin and then induced mesoderm differentiation. They found that CH223191 blocked the inhibitory effect of the drug, suggesting that AHR plays a critical role in the drug-induced toxicity pathway. This emphasizes also the potential use of these in vitro platforms to assess therapeutic strategies that could be implemented to avoid teratogenicity without compromising the intended therapeutic effect of the drugs.

Although the previously mentioned 2D cardiac differentiation platforms for heart developmental toxicity screening are simple, easy to implement and present a high accuracy in teratogenicity prediction, these models do not recapitulate tissue morphogenesis, thus compromising the capability of giving information regarding drug-induced morphological defects. Therefore, models that better recreate the tri-dimensional space of heart embryonic development may reveal additional and/or more insightful information regarding teratogenicity prediction. Simple 3D CM differentiation platforms and more in vivo-like heart organoid models are starting now to be explored in this field. Although 3D platforms for the directed differentiation of hPSCs into CMs [31,52] are able to recapitulate the diffusional gradients of molecules, a critical and more in vivo-like microenvironment parameter, they still lack the cellular structural organization. Since hPSC-derived heart organoids are only now being developed, the application of those models in heart developmental toxicity assays is still almost absent. Nevertheless, Hoang and co-workers took the first steps towards that direction by proving the applicability of a cardiac organoid model in the context of developmental cardiac toxicity. In this platform, hiPSC colonies were geometrically confined into 2D micropatterned surfaces and then, upon differentiation, the cells self-organized into 3D tissue-like structures, characterized by a core of CMs surrounded by a layer of smooth muscle-like cells. The authors selected nine drugs that could account for different teratogenic risks and then they evaluated the impact of continuous drug exposure until day 20 of differentiation, this being the time point at which they assessed different structural and functional parameters including CM area, 3D tissue morphology and contraction function [54].

Although this model allowed them to validate previously identified in vivo teratogenicity for the selected tested drugs, the relevance of using organoid models can be further explored if early and sequential time points throughout the organogenesis-like process are analyzed. Recently, Schmidt and co-workers briefly explored this idea as a proof-of-principle by using cardiac organoids that reflect different heart regions, namely outflow tract, atrioventricular canal, left and right ventricle and the atrial organoids [53]. Using a TNNI1-GFP reporter cell line, they assessed the impact of continuous drug exposure throughout the differentiation process, not only at the ending point of differentiation, opening in this way the path to understanding at which stage of development it is possible to induce defects in organogenesis. Additionally, since these organoids are a representation of the developmental process of different heart regions, this system may bring extra information regarding teratogenicity towards specific sub-populations of heart cells/regions.

## 7. Conclusions and Future Perspectives

As highlighted in this review, the number of women taking drugs during pregnancy has been increasing over the past decades. Therefore, the development of reliable models able to accurately predict the developmental toxicity of extant and newly developed drugs in a human context is mandatory. Since the heart is one of the first organs to be functional and is crucial for the survival of the fetus, cardiac developmental toxicity has been the focus of attention. So far, in vitro models, applied in developmental toxicity studies, have been able to corroborate and add insightful information regarding the teratogenicity of known drugs. However, there are still some challenges that should be taken into consideration in this field (Figure 1). Namely, it is important to understand if the studied drug can reach or cross the placenta. If so, the fraction of drug that can reach the bloodstream (bioavailability), and how the drug is metabolized by the human organism, must be determined. However, these relevant topics are still not taken into consideration in the majority of already reported in vitro developmental toxicity platforms and are crucial for prescribing safe and efficacious doses of orally administrated drugs. The development of micro-engineered models of the human placenta barrier, which normally recreate both an endothelium, based normally on HUVECs, and a trophoblastic-like epithelium, using primary human trophoblast cells to replicate the fetal–maternal interface of the placenta, respectively, have been already reported in the literature [55,56,57,58,59]. Additionally, although very little research has been explored so far on this issue, recent efforts have been made to establish human trophoblast models from hPSCs, with several reports already describing in vitro placenta models [60,61]. The integration of these placenta–barrier models with cardiac differentiating cells in a dynamic microfluidic device could be an interesting approach to better recreating the in vivo system and assessing in a more reliable way the impact of drug administration during pregnancy. These integrated systems may also be valuable to predict teratogenic drug dosage in a way that is more translatable to humans. Additionally, knowing that drugs may be safe in their native chemical configuration, but may produce toxic metabolites upon metabolization, we must develop models capable of predicting how a drug is metabolized in the human organism and/or when introduced to multi-organ systems, including in vitro models that recreate the liver, being the first-pass metabolization barrier, can assist in that prediction. Improved knowledge regarding transplacental drug transfer and metabolization will result in more reliable and translatable data to humans. Understand how drugs interact with placental transporters may also provide valuable information that can be explored in the future in the drug developmental design process in order to control the degree of drug interaction with the fetus. The use of targeted drug delivery systems, such as through the use of nanoparticles, may be an interesting and promising solution to deliver life-saving drugs to pregnant women while limiting fetal exposure [62].

Another debated question in this field is “Are organoid models relevant in developmental toxicology assays?” The introduction of heart organoid models in the context of developmental toxicity studies has been barely explored in the literature yet, and their relevance compared with more simplistic well-established hPSC-derived platforms has been questioned. Models that recreate organogenesis in vitro have the straightforward advantage of potentially allowing researchers to disclose tissue morphogenic defects, which is not feasible when using more simplistic models. However, these models bring additional challenges that cannot be disregarded, namely the intrinsic variability that is often linked with organoids and the potential challenge of incorporating these models in high-throughput settings [63]. Therefore, organoid models should be seen as a complementary test to current well-established and simpler 2D and 3D platforms [64,65,66,67], which are more suitable for assessing teratogenicity at the cell differentiation level and not specifically to revealing structural/morphological defects. Very recently described heart organoid/gastruloid models may be a valuable asset in this context [66,68,69,70].

The choice of a specific model, as well as the selected readouts, for developmental toxicity screening must take into consideration the aim of the study and the questions behind it. Independently of the selected model, variability in differentiation protocols and across different hPSC lines have been identified as key points that can easily compromise the accuracy of the platform in predicting teratogenicity [38,43,50]. Therefore, it is critical to establish robust differentiation models that can be also easily adapted and optimized for each hPSC line to avoid losing the predictive capacity. Additionally, standardized hPSCs passages and quality-control assays of pluripotency should be accounted for to avoid batch-to-batch variability. The selected readouts should be adapted to high-throughput screening environments when considering a first unbiased screening of several drugs, or be more informative when the intent is to disclose the mechanisms behind the observed toxicity. In the first case, the use of reporter cell lines associated with relevant biomarkers and fluorescent quantification, or brightfield images or video acquisition to account for morphological changes, are viable tools, allowing researchers to follow the progression of differentiation in real time. When the objective is to disclose the mechanisms behind toxicity, combined multiomics analysis will be key. Nevertheless, it is important to complement transcriptomic readouts with an assessment of specific functional features since transcriptome changes may not directly translate to significant physiological modifications.

Although the application of human-based in vitro models for developmental toxicity screening still has room for progression, the already-reported studies proved that these platforms can be a valuable asset in pre-clinical phase of drug testing and can be key in the future for human teratogenicity prediction.

## Figures and Tables

**Figure 1 ijms-24-04857-f001:**
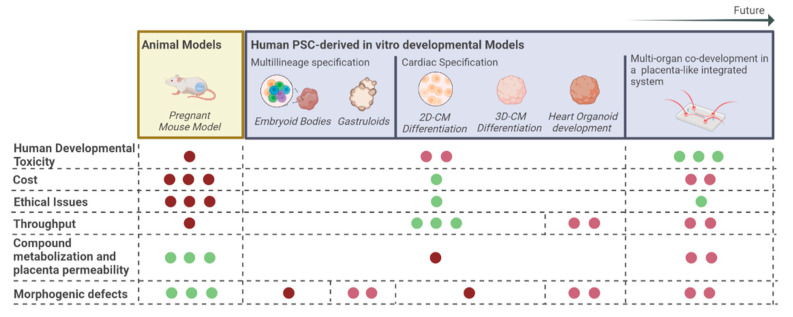
Comparison of animal models and in vitro hPSC-derived platforms to assess developmental toxicity. Legend: CM, Cardiomyocyte; ● low; ●● medium; ●●● high; red, negative issue; green, strong positive issue; pink, moderate positive issue.

**Table 1 ijms-24-04857-t001:** Humanized PSC-derived EBs and gastruloid models used to assess multilineage specification defects upon drug exposure. Legend: D, Day of differentiation; KO, Knockout; ULA, Ultra-Low Attachment; MACS, Magnetic-Activated Cell Sorting; Small molecules and growth factors: CHIR, GSK-3α/β inhibitor; SB431542, TGF-β1 Receptor ALK5 Inhibitor; Culture medium: DMEM-F12, E6; Commercially available microwell plates: AggreWell^TM^ plate.

MULTILINEAGE DIFFERENTIATION			
Reference	Drug(s)/Pollutants	Differentiation and Drug Exposure Strategy	Redouts
**Embryoid Bodies**			
**[37]**	Thalidomide	Culture Platform: V-shaped plate (D0–D4) + Bacteriological plate suspension culture (D4–D12) Duration of Differentiation: 14 days Drug Exposure: every other day D0 to D14 Differentiation Protocol: DMEM-F12 + 20% KO serum replacement + 1% non-essential amino acids + 0.1 mM β-mercaptoethanol	Transcriptomic profile: Microarray (D14)
**[35]**	Valproic acid	Culture Platform: V-shaped plate (D0–D4) + ULA plate (D4–D14) Duration of Differentiation: 14 days Drug Exposure: every other day from D0 to D14 Differentiation Protocol: DMEM-F12 + 20% KO serum replacement + 1% non-essential amino acids	Transcriptomic profile: RNA-sequencing (D14)
**[34]**	caffeine penicillin-G valproic acid	Culture platform: AggreWell plate (D0–D3) + 24-ULA plate (D13–D12) * Duration of Differentiation: 12 days Drug Exposure: every three days from D0 to D12 Differentiation Protocol: AggreWell™ EB Formation Medium * Use of MACS purified TRA-1-60 positive hPSCs	- EBs size and shape (D3, D6, D9, D12): BF images and circularity coefficient calculation - Mesoderm and ectoderm genes expression (D12): RT-PCR for the mesodermal genes KDR, C-ACTIN and BRACHYURY; and for the ectodermal genes NETO2, NCAM, NES, BIII-TUB and NEFH
**[39]**	Nicotine	Culture Platform: Aggrewell 800 plate (D0–D1) + ULA-6well plate (D1–D21) Duration of Differentiation: 21 days Drug Exposure: every day from D0 to D21 Differentiation Protocol: DMEM/F12 + 20% FBS + 2 mM L-glutamine + 1x non-essential amino acids + 100 mM β-mercaptoethanol	- Transcriptomic profile: Single-cell RNA sequencing (D21) - EBs size: BF images - ATP Activity: CellTiter-Glo 2.0 - ROS production: ROS-Glo H2O2
**[38]**	Valproic Acid All-trans Retinoic Acid Thalidomide Methotrexate Hydroxyurea Ascorbic acid Penicillin G Ibuprofen	Culture Platform: 96-Well Plate Duration of Differentiation: 7 days Drug Exposure: drug treatment on D0, D3 and D5 Differentiation Protocol: DMEM + 20% KO-SR + 1% GlutaMAX + 1% non-essential aminoacids + 0.18% β-mercaptoethanol,	Expression of 96 pre-selected developmentally genes: RT-PCR
**[36]**	folic acid all-trans retinoic acid dexamethasone valproic acid	Culture Platform: ULA plate (D0–D5) + 2D Matrigel (D5–D15) Duration of Differentiation: 15 days Drug exposure: every two days from D0 to D15 Differentiation Protocol: mTeSR1	Transcriptomic profile: RNA-sequencing (D15)
**[40]**	Cadmium	Culture Platform: 384-well plate Culture Duration: 8 days Drug exposure: every other day from D1 to D8 Differentiation Protocol: E6 medium	- Percentage of cTnT positive cells (D8): Flow Cytometry - Gene expression (D8)
**Gastruloids**			
**[41]**	All-*trans* retinoic acid Valproic Acid Bosentan Thalidomide Phenytoin Ibuprofen Penicillin G	Culture format: 96-Well plate Culture Duration: 72 h Drug exposure: 0–24 h Differentiation Protocol: 0–24 h: CHIR	- Morphological shape descriptors (circularity, lack of elongation, size): BF images - Fluorescence analysis: use of ES report cell line RUES2- GLR (mCit–*SOX2—neuroectoderm*, mCerulean–*BRA—mesoderm*, tdTomato–*SOX17—endoderm*)
**[42]**	Remdesivir	Culture format: 96-well plate Culture Duration: 5 days Drug exposure: single addition at D0 without medium change Differentiation Protocol: CHIR + SB431542 + retinoic acid	- Gastruloid morphologic analyses (area, elongation distortion index, and aspect ratio) (D5): BF images - Gene expression (D5): RT-PCR for the somites genes MEOX1, MESP2, PAX3); the cranial caudal axis genes ALDH1A2, FGF8, HOXB7, HOXB9, WNT5A; and for the neuroectoderm genes NEUROG2, OLIG3, PAX6

**Table 2 ijms-24-04857-t002:** Humanized PSC-derived 2D and 3D platforms used to assess mesendoderm and cardiac specification defects upon drug exposure. Legend: D, Day of differentiation; FBS, Fetal Bovine Serum; ULA, Ultra-Low Attachment; Small molecules and growth factors: CHIR, GSK-3α/β inhibitor; Activin A, activate the SMAD2/3 pathway of TFG-β signaling; BMP4, ligand of the TGF-β signaling; FGF2, fibroblast growth factor (FGF) that binds to FGF receptors and induces downstream signaling pathways; IWR-1, WNT-C59, IWP2 and XAV939, inhibitors of the WNT/β-catenin; Culture medium: RPMI.

MESENDODERM/CARDIAC DIFFERENTIATION			
Reference	Drug(s)/Pollutants	Differentiation and Drug Exposure Strategy	Redouts
**2D Culture**			
**[43]**	71 drug-like compounds	Differentiation Type: Mesendoderm commitment Culture Platform: 96-well Plate Duration of Differentiation: 3 days Drug Exposure: D0 and D1 Differentiation Protocol: D0–D1: RPMI + 2 mM L-glutamine + Activin A + WNT3A D1–D3: RPMI + L-glutamine + Activin A + 0.1% FBS	SOX17 protein expression (D3): Antibody staining (Counting the number of SOX17+ nuclei within the total DAPI+ nuclei)
**[44,45]**	30 drug-like compounds	Differentiation Type: Mesoderm commitment Culture Platform: micropatterned circular molds Duration of Differentiation: 3 days Drug Exposure: Single administration at D0(?) Differentiation Protocol: D0–D3: Activin A, BMP4, FGF2	Brachyury (T) protein expression (D3): Antibody staining (4 analyzed parameters: area of the T^+^ region; relative distance of the T^+^ region to the colony centroid and outline; standard deviation of the distribution of the T^+^ region; coefficient of variation of the distribution of the T^+^ region)
**[46]**	13-*cis*-retinoic acid (Isotretinoin)	Differentiation Type: CM differentiation Culture Platform: 12-well plate Duration of Differentiation: 6 days Drug Exposure: D0–D6 Differentiation Protocol: Gibco™ PSC Cardiomyocyte Differentiation Kit	Transcriptomic profile: RNA sequencing and ATAC sequencing (D0, D2 and D6):
**[47]**	Arsenic Trioxide	Differentiation Type: CM differentiation Culture Platform: Matrigel-coated plates Duration of Differentiation: >4 days Drug Exposure: compound was added at D0–D1 or D0–D2 Differentiation Protocol: D0–D2: differentiation basal medium I (CELLAPY) D2–D4: differentiation basal medium II (CELLAPY) D4-: differentiation basal medium III (CELLAPY)	- Cell Death: TUNEL assay (24 h of drug exposure) (D2, D4, D6) - Cell Proliferation: EdU staining (24 h of drug exposure) (D2, D4, D6) - Gene expression: RT-PCR for EOMES and T (24 h and 48 h of drug exposure) (D2); GATA4, MESP1, TBX5 (24 h of drug exposure) (D4); and α-ACTININ (24 h and 48 h of drug exposure) (D6?) - DNA damage: Immunostaining for γH2AX (24 h of drug exposure) (D2)
**[48]**	Dioxin	Differentiation Type: CM differentiation Culture Platform: Matrigel-coated plates Duration of Differentiation: 14 days Drug Exposure: From D-3 to D12 of differentiation (every time that the medium was changed) (Tested different exposure setups D-3–D0; D1–D4; D5–D8; D9–D14; D–3–D14) Differentiation Protocol: D0–D2: CHIR D2–D4: IWR-1	First assessment for all 5 tested conditions: - Gene expression (D14): RT-PCR for TNNT2, ACTN2 and MYL2 - Immunostaining for ACTN2 (D14) Drug exposure from (D-3–D0) - Gene expression: RT-PCR for T and GSC (D2); ISL1 and TBX5 (D5); TNNT2 and NKX2.5 (D8); ACTN2 and TNNT2 (D14) 2. Immunostaining for T (D2), ISL1 (D5), TNNT2 and ACTN2 (D8, D14) 3. Transcriptomic profile (D2): RNA-sequencing
**[49]**	Ribavirin	Differentiation Type: CM differentiation Culture Platform: Matrigel-coated plates Duration of Differentiation: 7 days Drug Exposure: D0, D1, D3, D5 (Tested different drug exposure setups: From 1. hPSC to mesoderm; 2. mesoderm to cardiac progenitors; 3. cardiac progenitors to cardiomyocytes) Differentiation Protocol: D0–D1: CHIR D3–D5: IWR-1	- Contraction (visual beat score): Frequency of contraction and beating colonies - Cell Proliferation: EdU staining - ROS content: DCFH-DA staining - Gene expression: RT-PCR for EOMES and T (mesoderm); GATA4 and ISL1 (cardiac progenitors); cTnT and α-MHC (cardiomyocytes)
**[50]**	Drugs used for platform validation: Thalidomide Valproic acid Folic acid Saccharin Additional 17 drugs for platform testing	Differentiation Type: CM differentiation Culture Platform: Matrigel-coated 24-well plates Duration of Differentiation: 14 days Drug Exposure: D0, D3, D7, D10 Differentiation Protocol: D0–D3: CHIR + BMP4 + ActivinA D3–D5: XAV939	- Primary redout: gene expression of BMP4 (D7) and MYH6 (D14) - Secondary redout (only used to further support the first one): Contraction (visual beat score)
**[40]**	Cadmium	Differentiation Type: CM differentiation Culture Platform: Matrigel-coated plates Duration of Differentiation: 8 days Drug Exposure: Every other day from D1 to D8 (Tested different drug exposure setups (D0–D2 or D2–D4)) Differentiation Protocol: STEMdiff^TM^ Cardiomyocyte Differentiation Kit	- NKX2.5-positive cells (D8): Flow Cytometry (NKX2.5-GFP reporter cell line) - Gene expression: RT-PCR for MESP1, EOMES, MIXL1, HAND1, SNAI2 and HOPX (D2); and NKX2–5, GATA4, TNNT2, α-actinin (D8) - Histone methylation (H3K27 trimethylation (H3K27me3) and H3K4 trimethylation (H3K4me3)) (D2): Western Blot
**[51]**	5-Fluorouracil Penicillin G	Differentiation Type: CM differentiation Culture Platform: Matrigel-coated 24-well plates Duration of Differentiation: 10 days Drug Exposure: D2, D3, D4, D6, and D8 Differentiation Protocol: D0–D1: CHIR + BMP4 + FGF2 D2–D4: IWP2	- Contraction (D10): BF video (frequency of contraction and contraction area) - Gene expression (D10): RT-PCR for TNNT2 and ACTN2
**3D Culture**			
**[31]**	Thalidomide Valproic acid Epoxiconazole	Model: CM aggregates Culture Platform: ULA 96-well plate Duration of Differentiation: 7 Days Drug Exposure: D1, D2, D3, D6 (do not expose at D0) Differentiation Protocol: D0–D1: CHIR + BMP4 + ActivinA + FGF2 D2–D3: WNT-C59	- Contraction (D7): Visual beat score (if no movement was seen, the beat score 0 was given; if the entire area of the sphere contracted, a beat score of 2 was given; everything in between was given a score of 1) - Aggregate size (D7): BF images
**[52]**	Thalidomide Valproic acid	Model: CM aggregates Culture Platform: ULA 96-well plate Duration of Differentiation: 7 Days Drug Exposure: D1, D2, D3, D6 (do not expose at D0) Differentiation Protocol: D0–D1: CHIR + BMP4 + ActivinA + FGF2 D2–D3: WNT-C59	- Contraction (D7): BF video - NKX2.5 expression level (D7): Luminescence measurement using a NKX2.5-reporter line
**[54]**	Doxylamine Succinate Amoxicillin Rifampicin Lithium Carbonate Phenytoin Doxycycline All-trans-RA (Tretinoin) 13-cis-RA (Isotretinoin) Thalidomide	Model: Cardiac Organoid Culture Platform: Micropatterned platform Duration of Differentiation: 20 Days Drug Exposure: D1, D2, D4, D6–D20 (do not expose at D0) Differentiation Protocol: D0–D1: CHIR D2–D4: IWP4	- Percentage of cTnT-positive cells (D20): Flow cytometry - cTnT-positive area (D20): Area ratio - 3D tissue morphology (D20) - Contraction (D20): BF video and calcium transient profile (hPSC line engineered with a calcium sensor GCaMP6f) (Contraction velocity and BPM)
**[53]**	Aspirin Thalidomide Retinoid Plastic residuals	Model: AVC; LV and RV. OFT, atrial organoids Culture Platform: 96-well plate Duration of differentiation: 9.5 days	- Gene expression (D3.5, D4.5 and D9.5) - Morphology (D3.5, D4.5 and D9.5): BF image - Patterning (D3.5, D4.5 and D9.5): TNNI1-GFP reporter cell line
**[40]**	Cadmium	Model: Cardiac Organoids (atrial and ventricle CMs, CFs, endocardial and endothelial cells) Culture Platform: Culture Duration: 8 days Drug Exposure: every other day from D1 to D8 Differentiation Protocol: D0–D1: CHIR + BMP4 + Activin A D2–D4: IWR-1	- Contraction (D8) - Gene expression (D8): RT-PCR for selected genes associated with cardiomyocytes, endocardial cells, and cardiac fibroblasts

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
