# Peer review of "Developmental Toxicity Studies: The Path towards Humanized 3D Stem Cell-Based Models"

_ijms, 2023, doi:10.3390/ijms24054857_

Round 1
Author Response
The review by Branco et al is a timely and very nice overview of alternative ways of assessing teratogenic effects with a focus on stem cell models. The review is well written.
We thank Reviewer 1 for taking the time to read our manuscript and for the constructive suggestions given. Please find below a point-by-point response to all comments.
No figures are included in this paper, and one or two good overview figures would be appreciated and would greatly improve the review. For instance, a figure on the gastrulation process compared to cardiomyocyte differentiation from EBs.
We agree with the reviewer in that this review would benefit from having additional figures. Thus, in addition to the graphical abstract that was already present in the first version of the manuscript, we added one figure that gives an overview of the different in vitro hPSC-derived platforms that have been used in developmental toxicity assays, by making a side-by-side comparison with in vivo animal models.
Layout of Table 1 and 2 is too bad (line distance, landscape better?), so please improve the layout of the tables.
We agree with the reviewer that the presentation of these tables is not the most readable one. Thus, we improved the layout of Tables 1 and 2 accordingly.
Your major focus is on drugs, but what about environmental chemicals? The human fetus is exposed to many environmental chemicals (phthalates, bisphenols, POPs, PFAS etc) and many of these chemicals can be measured in human cord blood, which serves as a good proxy for fetal exposure. Please add a paragraph on this issue and the need for NAMs to test for development toxicity of these compounds as well.
We thank the Reviewer for this pertinent observation. To account for review’s comment, we added a paragraph in the introduction section (Page 2, Line 82-87): “Apart from drug administration, environmental pollution exposure during pregnancy has been also considered as a critical point that should be carefully accounted. In fact, maternal exposure to metals, chemicals, and toxins, and their link to congenital anomalies, is well documented (reviewed in [1]), with the World Health Organization estimating that 5% (2—10%) of all congenital anomalies are attributable to environmental causes [2].”
Moreover, we would also like to highlight that the previous version of our manuscript already included a revision of developmental toxicity studies that tested the impact of chemicals/pollutants using hPSC-derived models, namely, Wu et al, 2022; Ye et al., 2020 and Schmidt et al., 2022 studies.
In order to evaluate the relevance of the 3D cardiomyocyte differentiation model from EBs, it might be worthwhile to look at this reference, in which biomarkers of hiPSC have been compared to biomarkers in pig and murine embryos: LAUSCHKE K, VOLPINI L, LIU Y, VINGGAARD AM, HALL VJ. A comparative assessment of marker expression between cardiomyocyte differentiation of human iPSCs and the developing pig heart. Stem Cells and Development 30 (7), 374-385, 2021.
We thank the reviewer for highlighting this point. We agree with the reviewer in that the knowledge provided by the mentioned study, and other similar studies, may be useful to help selecting the right biomarkers to help improving differentiation of hiPSC towards more in vivo-like cardiac tissue in the future. Nevertheless, we believe that this type of studies would be more relevant if performed using 3D cardiomyocyte directed differentiation platforms instead of EBs. In the case of EBs, being an unbiased differentiation protocol, the specification of cardiomyocytes is not controlled and therefore their application is more relevant when assessing the impact of a substance at the early stages of the three germ layers specification, meaning at the multilineage level. When the focus is on understanding the specific impact of a substance on normal cardiac cell’s specification and heart development, directed 3D differentiation protocols are more robust models and will resemble better the heart organogenesis process. Additionally, with 3D cardiac differentiation platforms it is possible to assess structural defects and, through signalling pathways modulation, assess the impact of a compound specifically on subpopulations of cardiac cells (left and right ventricle cardiomyocytes; atrial cardiomyocytes, …).
The statement: ‘Unlike EBs, which are 3D models of spontaneous and unbiased differentiation of PSCs that do not present any in vivo-like structural organization……’ is too much a black/white statement. The EBs represent the human blastocyst and investigation of any toxic effect of compounds on the blastocyst in itself are also relevant. Please rephrase.
We totally recognize the relevance of EBs in this field, being irrefutably that they represent a viable strategy to assess human developmental toxicity in vitro. In fact, we showed and highlighted in the manuscript, with different studies, that these models allowed to attain relevant information regarding the impact that drugs/chemical/pollutants have on normal germ layers, and their derivatives, specification. However, the mentioned statement intends only to make clear the difference between EBs and gastruloid models. In EBs it is not possible to observe any structural organization among the different cell types, contrarily to what is observed in gastruloids, where there is an in-vivo like patterning, showing anteroposterior axial elongation. We do not intend to claim that one model is better than the other, but instead that, with gastruloids, we can have extra information regarding spatial defects. Therefore, we think that the mentioned statement does not question the relevance of EBs in this field. Nevertheless, to make clear that we recognize the relevance of the EB’s model, we modified the paragraph where the statement is included (Page 5; Lines 203-207): “Although EBs reflect cellular differentiation into all three germ layers, and therefore are relevant models to study the toxic effect of compounds at the blastocyst stage, gastruloids, being a model system that mimics some of the events of gastrulation, including symmetry break and axial elongation, have the additional asset of potentially giving information regarding spatial organization defects.
This comment also applies to this statement: L. 347: ‘Although 3D platforms for directed differentiation of hPSCs into CMs [30][49] are able to recapitulate the diffusional gradients of molecules, a critical and more in-vivo like microenvironment parameter, they still lack the cellular structural organization’. This could be an argument for the inclusion of a panel of assays to test for developmental toxicity, which each represent various stages of embryonic development.
We agree with the reviewer that more simplistic models can be as interesting or as relevant as more complex ones. In fact, in the “conclusion and future perspectives” section, we mentioned that “Models that recreate organogenesis in vitro have a straightforward advantage of potentially allowing to disclose tissue morphogenic defects, which is not feasible when using more simplistic models. However, these models bring additional challenges that cannot be disregarded, namely the intrinsic variability that is often linked with organoids and the potential challenge of incorporating these models in high-throughput settings [60]. Therefore, organoid models should be seen as a complementary test to current well-established and simpler 2D and 3D platforms, which are more suitable for assessing teratogenicity at the cell differentiation level and not specifically to reveal structural/morphological defects.”. We think that this paragraph is in line with the reviewer’s comment, specifically that we should consider a panel of assays to test for developmental toxicity in vitro, and that the selected models should be the most relevant ones to answer a specific biological question.
Minor
L 161: ‘Although in some animal models, moderate congenital malformations have been described upon thalidomide exposure, in mice it was not observed significant fetal changes’: Please rephrase sentence.
We followed the suggestion made by the reviewer and rephrased the sentence (Page. 4; Lines 162-169): “Although animals or animal-based in vitro models have been essential for toxicology testing in the past years, species differences between humans and animals have been responsible for the lack of drug-induced teratogenicity detection. The most known and recognized case is the “Thalidomide Tragedy”, in which the teratogenicity of this drug was not foreseen in mouse models leading to numerous embryonic, fetal and neonatal deaths, and severe congenital malformations in humans.”
L. 164: Concerning the statement: ‘only 70–80% of the teratogenic cases observed in rats and rabbits are reflected in humans [30],….’: Please refer to the original reference: Daston GP, Knudsen TB (2010) 12.02—Fundamental concepts, current regulatory design and interpretation. Comprehensive toxicology. Elsevier, Amsterdam, pp 3–9
We thank the reviewer for the suggestion. We changed the text to refer the original reference
L. 184: ’focus 1)….’: Please insert a ‘focus on 1)….’
We corrected this typo according to the reviewer suggestion.
L. 191: [39][39]: Please delete the one [39]
We deleted the duplicated reference according to the reviewer suggestion.
L. 238: ‘this methodology is not compatible when a screening of a set of different drugs is needed in the first place’: Please rephrase this sentence
We followed the suggestion made by the reviewer and rephrased the sentence accordingly (Page 5; Lines 245-248): “Although the advantages of using RNA-seq analysis to deeply understand the pathways behind the observed developmental toxicity are irrefutable, this methodology is not compatible with high throughput screening settings.”
L. 259: ’analysed trough’: Please correct typo
We corrected the typo according to the reviewer suggestion.
L. 324: ’organic polluting Dioxin’: Please call it a pollutant or contaminant
We changed the word polluting to pollutant according to the reviewer suggestion.
L. 423: ‘Therefore, organoid models should be seen as a complementary test to current well established and simpler 2D and 3D platforms’: Please provide references here both for organoid models and 2/3D models.
We followed the reviewer’s suggestion and added the references accordingly (Pag. 15; Lines 448-452)
Reviewer 2 Report
In this manuscript, the authors described the pathway toward the introduction of human pluripotent stem cell-derived models in developmental toxicity studies. The authors focused on 3D stem cell-based models that recapitulate two very important early developmental stages, gastrulation, and cardiac specification.
In the introduction part, the authors pointed out the teratogenic risk in humans is often missing for most of the purchased drugs. Meanwhile, medication use during pregnancy is very common. Therefore, it is really important to get development toxicity information for the drugs.
After that, the authors summarized current animal models and in vitro animal-based models in developmental toxicity assessment. They pointed out that these models still have a lot of limitations in assessing developmental toxicity.
Then, the authors summarized in vitro humanized models to assess developmental toxicity with an emphasis on the use of hPSC-derived EBs and gastruloids for multilineage developmental toxicity studies and the use of hPSC-derived 2D and 3D models for cardiac developmental toxicity studies.
The language used in the article is clear and professional. However, some of the citation formats can be improved.
Line 68, the citation of 14, 13, 15 can be re-ordered, similar problem in line 135 and 272
The format for citing multiple articles should be synchronized. Line 190 used [32-37], but line 395 and 427 itemized every single article. Such problems are across the whole article.
It would be better if the authors can provide an illustration of different phases of human embryo development to show the timeline of different toxicity effects. Also, the authors mentioned a lot of 3D culturing models and mediums in Table 1 and 2. It would be better if the authors can give some background information and illustration about such topic. In this part, a lot of the terminology is not explained well to the readers. For example, CHIR.
In summary, this review is a good information source for in vitro models development and gives the scientific community an update on this topic.
Author Response
In this manuscript, the authors described the pathway toward the introduction of human pluripotent stem cell-derived models in developmental toxicity studies. The authors focused on 3D stem cell-based models that recapitulate two very important early developmental stages, gastrulation, and cardiac specification.
In the introduction part, the authors pointed out the teratogenic risk in humans is often missing for most of the purchased drugs. Meanwhile, medication use during pregnancy is very common. Therefore, it is really important to get development toxicity information for the drugs.
After that, the authors summarized current animal models and in vitro animal-based models in developmental toxicity assessment. They pointed out that these models still have a lot of limitations in assessing developmental toxicity.
Then, the authors summarized in vitro humanized models to assess developmental toxicity with an emphasis on the use of hPSC-derived EBs and gastruloids for multilineage developmental toxicity studies and the use of hPSC-derived 2D and 3D models for cardiac developmental toxicity studies.
The language used in the article is clear and professional. However, some of the citation formats can be improved.
We thank Reviewer 2 for taking the time to read our manuscript and for the constructive suggestions given. Please find below a point-by-point response to all comments.
Line 68, the citation of 14, 13, 15 can be re-ordered, similar problem in line 135 and 272
The format for citing multiple articles should be synchronized. Line 190 used [32-37], but line 395 and 427 itemized every single article. Such problems are across the whole article.
We thank the reviewer for highlighting this problem. We checked for all the cases where multiple articles citations were not well presented and pointed for editorial formatting.
It would be better if the authors can provide an illustration of different phases of human embryo development to show the timeline of different toxicity effects. Also, the authors mentioned a lot of 3D culturing models and mediums in Table 1 and 2. It would be better if the authors can give some background information and illustration about such topic. In this part, a lot of the terminology is not explained well to the readers. For example, CHIR.
We agree with the reviewer, and, in addition to the graphical abstract, we added one figure that gives an overview of the different in vitro hPSC-derived platforms that have been used in developmental toxicity assays, making a side-by-side comparison with in vivo animal models.
Also, in Tables’ legend we provided some explanations about the terminology used.
In summary, this review is a good information source for in vitro models development and gives the scientific community an update on this topic.
Round 2
Reviewer 1 Report
The paper can be accepted now. Best regards